# Saturated free fatty acids and association with memory formation

Tristan P. Wallis[1,6], Bharat G. Venkatesh[1,6], Vinod K. Narayana [1,4,6], David Kvaskoff[1,5], Alan Ho[2], Robert K. Sullivan [2], François Windels [2], Pankaj Sah [2,3] & Frédéric A. Meunier [1✉]

Polyunsaturated free fatty acids (FFAs) such as arachidonic acid, released by phospholipase activity on membrane phospholipids, have long been considered beneficial for learning and memory and are known modulators of neurotransmission and synaptic plasticity. However, the precise nature of other FFA and phospholipid changes in specific areas of the brain during learning is unknown. Here, using a targeted lipidomics approach to characterise FFAs and phospholipids across the rat brain, we demonstrated that the highest concentrations of these analytes were found in areas of the brain classically involved in fear learning and memory, such as the amygdala. Auditory fear conditioning led to an increase in saturated (particularly myristic and palmitic acids) and to a lesser extent unsaturated FFAs (predominantly arachidonic acid) in the amygdala and prefrontal cortex. Both fear conditioning and changes in FFA required activation of NMDA receptors. These results suggest a role for saturated FFAs in memory acquisition.

[1] Clem Jones Centre for Ageing Dementia Research, Queensland Brain Institute, The University of Queensland, St Lucia, QLD, Australia. [2] Queensland Brain Institute, The University of Queensland, St Lucia, QLD, Australia. [3] Joint Center for Neuroscience and Neural Engineering, and Department of Biology, Southern University of Science and Technology, Shenzhen, Guangdong Province, P. R. China. [4] Present address: Metabolomics Australia, Bio21 Institute, The University of Melbourne, Melbourne, VIC, Australia. [5] Present address: Boehringer Ingelheim Pharma GmbH & Co. KG, Drug Discovery Sciences, Biberach an der Riß, Germany. [6] These authors contributed equally: Tristan P. Wallis, Bharat G. Venkatesh, Vinod K. Narayana. ✉email: f.meunier@uq.edu.au

Elucidating the cellular mechanisms that mediate learning and memory formation remains one of the key challenges in neurobiology. Studies over many decades have established that a host of protein and gene responses contribute to learning and memory acquisition[1]. However, despite the clear abundance of phospholipids and their metabolites in the brain, and their established roles in neural signalling, surprisingly little is known about their contribution to these important processes. Lipids have multiple structural, signalling, developmental and metabolic functions in the brain[2]. In particular, phospholipids are crucial components of the neuronal plasma and synaptic vesicle membranes, and are therefore considered to be essential for neurotransmission, synaptic plasticity and memory formation[3–8]. The vesicular trafficking underpinning these processes involves tightly regulated dynamic modulation of phospholipid membrane fluidity, curvature and surface chemistry in concert with protein/protein and protein/lipid interactions at the pre- and post-synapse[9,10]. These processes are mediated in part by the action of specific phospholipases that can locally modify the phospholipid landscape[11,12]. This enzymatic phospholipid remodelling generates phospholipid metabolites such as diacylglycerols, inositol triphosphate, lysophospholipids and free fatty acids (FFAs), which can affect membrane dynamics[13,14] and act as lipid signalling molecules[15–18]. In particular, protein lipidation such as myristoylation and palmitoylation, involving the transfer of saturated myristic and palmitic acids to synaptic proteins, is preponderant in the nervous system[19] and critical for synaptic plasticity[20–26]. Investigation of the roles of phospholipid metabolites in regulating key processes in memory formation is therefore an essential complement to protein, gene and structural studies in this area.

The majority of the phospholipids in neuronal membranes are ester-linked glycerophospholipids. Fatty acids represent the two hydrophobic tails at the sn-1 and sn-2 positions of the phospholipid glycerol backbone, with a hydrophilic sn-3 head group defining the phospholipid class[27]. Canonical phospholipids are generally considered to be enriched with saturated fatty acids such as palmitic acid (C16:0) and stearic acid (C18:0) at the sn-1 position and unsaturated fatty acids such as arachidonic acid (C20:4) and docosahexaenoic acid (DHA, C22:6) at the sn-2 position[28]. Unsaturated fatty acids have long been considered important to health, and early studies correlated a reduction in brain polyunsaturated fatty acids with ageing and memory impairment[29]. Multiple neuronal processes mediated by phospholipase A2 (PLA$_2$) lead to the release of unsaturated FFAs (particularly arachidonic acid) from the sn-2 position of specific phospholipids. Arachidonic acid has been demonstrated to variously modify neurotransmitter release[30–34], modulate membrane fluidity[14], influence long-term potentiation (LTP) via its ability to diffuse through the synaptic cleft[35,36], regulate synaptic transmission either alone or in combination with other FFAs and lysophospholipids[35,37], and initiate inflammation[38]. The status of arachidonic acid as a key mediator of neurotransmission and LTP is, however, at odds with its surprisingly low abundance relative to other FFAs in neurons. Indeed, using sensitive isotope labelling multiplex analysis of FFAs (FFAST)[39], we recently demonstrated that stimulation of both rat neurons and neurosecretory cells in vitro leads to the predominant generation of saturated FFAs, suggesting that these FFAs could also be involved in neurotransmission, learning and memory in the brain.

To explore this possibility, we used FFAST to map the distribution of 18 FFAs in the rat brain, and their changes in response to learning using auditory fear conditioning (AFC). We also applied a targeted LCMS lipidomic workflow to quantify the responses of 135 phospholipid species belonging to 5 different classes. This combined approach revealed that the highest phospholipid and FFA concentrations, relative to tissue weight, are in the amygdala and prefrontal cortex, and that these are dramatically modified by fear conditioning. These FFA changes were largely driven by long chain saturated FFAs such as myristic and palmitic acids, and were not seen when learning was inhibited by using the NMDA receptor antagonist CPP. Together these data suggest that phospholipid-derived FFA increases play an unexpected and critical role in learning and memory formation.

## Results

**Phospholipids and FFAs are non-homogeneously distributed across the rat brain.** For this study, we employed two complementary lipidomic techniques to examine changes in the phospholipid and FFA landscape in rat brain tissue. FFAST uses differential stable isotope tagging to label the carboxyl group of FFAs, thereby greatly increasing the ionization efficiency and sensitivity and allowing multiplexing to minimize analytical variance[39]. We took advantage of the nanomolar sensitivity of FFAST to quantitatively survey the FFA landscape (18 targeted FFA species, Supplementary Table 1) across 6 regions of the rat brain, reporting FFA abundances in the pmol/mg wet weight range consistent with previous literature[40]. For phospholipid analysis, we relied on diagnostic fragmentation using negative mode electrospray ionization mass spectrometry (ESI-MS) to generate a precursor ion/sn-1/sn-2 acyl fragment fingerprint for each phospholipid species (Supplementary Table 1), such that we identified, for example, the sum composition PC34:1 as the molecular species PC18:1_16:0 (but not as positional isomer PC18:1|16:0 where 18:1 sn-1, 16:0 sn-2).

Following perfusion with ice-cold artificial cerebrospinal fluid to flush blood and chill the brain, brains were quickly removed from the skull and snap frozen in liquid nitrogen, sectioned using a cryostat, and tissue from thin sections was dissected and transferred into tubes on dry ice. These conditions were designed to minimize ischaemic and post-mortem lipid metabolism. FFAs and phospholipids were extracted from dissected tissue (central amygdala, basolateral amygdala, prefrontal cortex, ventral hippocampus, dorsal hippocampus and cerebellum) and independently analyzed (Fig. 1a, b). Stacked bar plot profiles representing the average concentrations (pmol analyte per mg of brain tissue) of the FFAs and 5 classes of phospholipids[41] (phosphatidic acid—PA, phosphatidylcholine—PC, phosphatidylethanolamine—PE, phosphatidylglycerol—PG and phosphatidylserine—PS) measured across eight animals were generated (Fig. 1d–i).

FFAs and phospholipids were both found to be non-homogenously distributed across the assayed brain regions, with the highest concentrations relative to tissue weight in the amygdala and prefrontal cortex, and lowest concentrations in the dorsal hippocampus and cerebellum. Saturated FFAs (6:0–24:0) constituted the majority of the FFAs across all brain regions. DHA, arachidonic acid and oleic acid (C18:1) were the predominantly observed unsaturated FFAs. The predominance of saturated FFAs is in agreement with our previous findings in cultured rat neurons and neurosecretory cells[39]. Each of the phospholipid classes exhibited a different distribution of acyl species, with 18:1_16:0 notably abundant. PE and PS were also characterized by high relative abundance of species containing DHA. Overall, PC and PE were present in by far the highest abundance across the brain.

**Phospholipids and FFAs respond to associative learning.** The remarkably high relative abundance of phospholipids and FFA in the amygdala suggested that they could be involved in aspects of learning and memory acquisition that engage the amygdala[42]. We

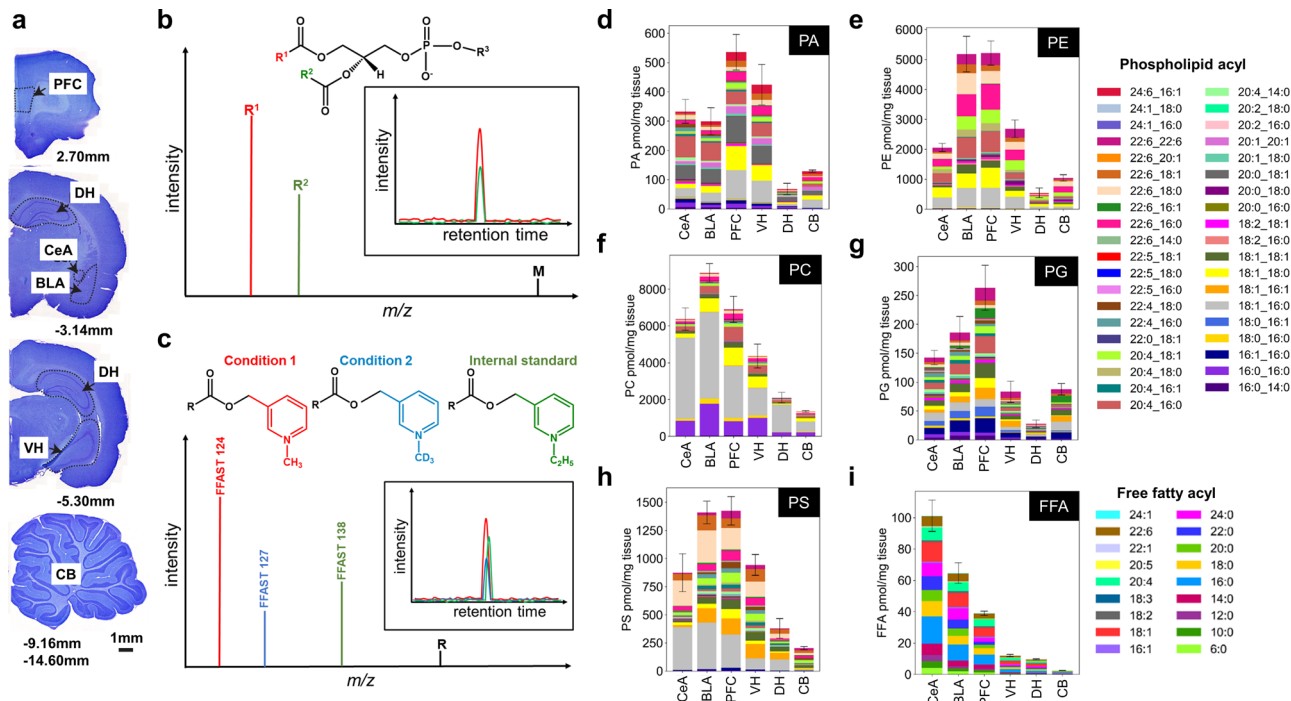

**Fig. 1 Distribution of free fatty acid (FFA) and phospholipid in the rat brain. a** For each of the 32 animals used in this study, tissue was dissected from the basolateral amygdala (BLA), central amygdala (CeA), the prefrontal cortex of the forebrain (PFC), ventral hippocampus (VH), dorsal hippocampus (DH) and cerebellum (CB), as indicated by dotted lines on Cresyl violet-stained brain sections with corresponding Bregma co-ordinates. Scale bar = 1 mm. **b** Identification and quantification of phospholipids using diagnostic ion fragmentation in liquid chromatography mass spectrometry (LCMS). Shown is a hypothetical phospholipid negative mode fragmentation mass spectrum. The parent (M), sn-1 (R1) and sn-2 (R2) fragment ion masses are unique to each species, and are used for multiple reaction monitoring (MRM, inset) LCMS to quantify abundance. **c** Schematic representation of FFA analysis using Free Fatty Acid Stable isotope Tagging (FFAST). For a given brain region the FFAs extracted from animals from different experimental conditions (saline paired, saline unpaired, CPP paired, CPP unpaired, see Fig. 2) were individually labelled at the carboxy-terminus using FFAST-124 or FFAST-127. Samples were combined, spiked with FFAST-138-labelled internal standards, and analyzed by LCMS. The 3 labelled variants of each FFA species display similar chromatographic elution times, and the ratio of each FFAST fragment relative to the internal standard fragment allows quantification of the abundance of the FFA in the condition. This workflow was repeated 8 times, to establish FFA abundance in each of the 8 animals used in each experimental condition. **d–i** Profile measurements of FFAs and 5 classes of phospholipids (PA—phosphatidic acid, PC—phosphatidylcholine, PE—phosphatidylethanolamine, PG—phosphatidylglycerol, PS—phosphatidylserine) across the brains of the control (saline unpaired) rats from auditory fear conditioning experiments, with analytes shown by acyl chain composition. Bars represent the total analyte measurement, with coloured sub-bars corresponding to the mean individual analyte concentrations (pmol/mg tissue) observed across 8 animals. Error bars represent the cumulative standard error of the mean (SEM) for all analytes. Source data are provided as a Source Data file.

explored this possibility by investigating the response of these lipids to AFC, a widely used learning paradigm in which animals learn to associate a neutral auditory tone with a mild electric footshock[43,44]. Rats exposed to paired tone/shock AFC stimuli (paired animals) exhibited significant freezing to a subsequent tone-only stimulus[43,44] (Fig. 2a, b). Control rats (unpaired animals) were exposed to the same number of unpaired tones. We sought to determine whether this long-term learned behaviour correlated with changes in the lipid landscape across the animals' brains, and more specifically of the amygdala. N-methyl-D-aspartate (NMDA) receptors are critical in fear learning and the competitive NMDA receptor antagonist (6)-3-(2-carboxypiperazin-4-yl)propyl-1-phosphonic acid (CPP)[45] is routinely used to block associative learning[44,46] (Supplementary Fig. 1). Accordingly, we tested whether blocking NMDA receptors with CPP altered the lipid response to AFC, compared to that in saline-injected animals. Comparison of the lipidomic profiles (Supplementary Fig. 2) of each of the four cohorts of animals (Fig. 2a) allowed us to evaluate the FFA and phospholipid response to AFC and how this activity-dependent response was modulated when long-term fear memory acquisition was blocked with CPP, thereby potentially identifying lipid species whose abundance changes correlate with NMDA-dependent long-term memory.

At a brain-wide level, paired AFC with saline injection resulted in significant increases in a number of FFA species, most notably saturated myristic acid, and significant decreases in a number of phospholipids (Fig. 2c). Strikingly, paired AFC in CPP conditions was characterized by decreases in both FFAs and phospholipids (Fig. 2d). In saline-treated animals, AFC drove a strong overall increase in FFAs, a slight increase in PG, and decreases in PA, PC, PE and PS (Fig. 2d). By contrast, in CPP-treated animals, AFC drove overall decreases in all classes. Examination of the absolute and fold change in abundance of all lipid analytes across all brain regions (Supplementary Fig. 3a, b) demonstrated the complexity of the overall lipid response to AFC. However, features revealed in the hierarchical clustering heatmap (Fig. 3a) did highlight various trends across regions: FFAs and phospholipids from a particular class tended to cluster together in the dendrogram, indicative of a similar response to AFC, although a given class of lipid could exhibit completely different responses to AFC in different brain regions. Deeper examination of these general trends revealed a number of specific activity-dependent lipid class changes, which were affected by CPP treatment: as noted above, FFAs tended to exhibit an increase across the brain, with the strongest effect being observed in the central amygdala (Fig. 3a, b, dotted boxes 1), and most strongly with myristic acid; PA and PC

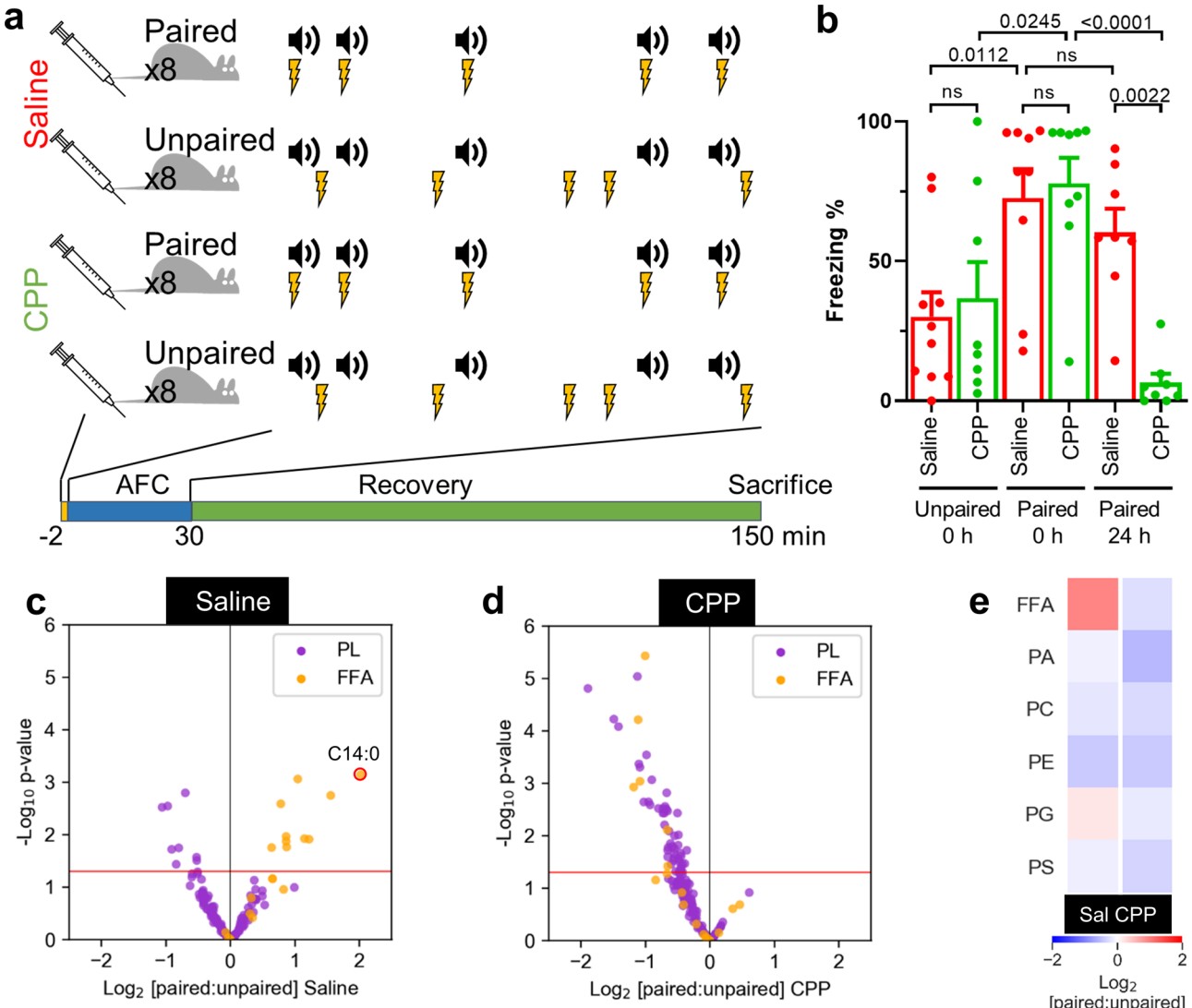

**Fig. 2 Behavioural and lipid responses to auditory fear conditioning (AFC) require N-methyl-D-aspartate (NMDA) receptors. a** Complete AFC workflow. Rats were injected intraperitoneally with either 0.9% sterile saline (red) or 10 mg/kg CPP ((6)-3-(2-carboxypiperazin-4-yl)propyl-1-phosphonic acid, green) and introduced to the AFC apparatus for 2 min of context habituation. Animals were exposed to either paired or unpaired auditory tones and footshock (cohorts of 8 animals for each of the 4 experimental conditions) for 30 min, and returned to their home cages for 2 h. Rats exposed to paired stimuli exhibit a subsequent freezing response to tone only stimulation, rats exposed to unpaired stimuli exhibit a reduced freezing response. **b** Typical effect of paired AFC measured by the percentage freezing response to subsequent tone only stimulus at 0 h and 24 h post AFC. From left to right, each cohort contained the following biologically independent animals: 10, 8, 9, 9, 8, 8. Error bars represent the standard error of the mean (SEM). The significance of the difference between freezing responses was determined by one-way ANOVA with Sidak multiple comparison, $p$ values < 0.05 are indicated, ns = not significant. Animals tested immediately following AFC paired stimuli exhibited an equivalent response to the tone in both the saline- (red) and CPP-treated (green) cohorts. For paired animals from an independent cohort tested 24 h subsequent to AFC, the CPP-treated animals exhibited significantly reduced response. **c**, **d** Average response of individual FFAs and phospholipids to paired AFC. Each dot on the volcano plot represents the average change in abundance of a single lipid analyte across 6 measured brain regions, in paired versus unpaired AFC in saline or CPP treated animals. Analytes below the red line represent those whose change in abundance was not statistically significant (two-tailed $t$-test $p$ > 0.05). Myristic acid (C14:0) is highlighted with a red circle. **e** Average response of each lipid class to paired AFC. Each pixel in the heatmap represents the average change in total abundance of all lipids of a given class across the 6 measured brain regions. PA—phosphatidic acid, PC—phosphatidylcholine, PE—phosphatidylethanolamine, PG—phosphatidylglycerol, PS—phosphatidylserine. Source data are provided as a Source Data file.

levels strongly decreased in the ventral hippocampus (Fig. 3a, b, dotted boxes 2) whereas PE levels decreased in the central and basolateral amygdala (Fig. 3a, b, dotted box 3) and PG levels increased in the ventral hippocampus (Fig. 3a, b, dotted boxes 4). All lipid classes increased slightly in the prefrontal cortex (Fig. 3b, dotted box 5). CPP generally modulated all of these responses, but the effect was particularly strong in the central amygdala and dorsal hippocampus, where a majority of the assayed lipids showed a very strong decrease in abundance. Although CPP itself

affected basal levels of FFAs across the brain (Supplementary Fig. 1b), these changes did not correlate with the region-specific CPP effects in response to activity.

**FFA responses correlate with formation of long-term fear memory.** As shown in Figs. 2 and 3, FFAs (particularly saturated species such as myristic and palmitic acids) exhibited the most striking and consistent activity-dependent changes in abundance

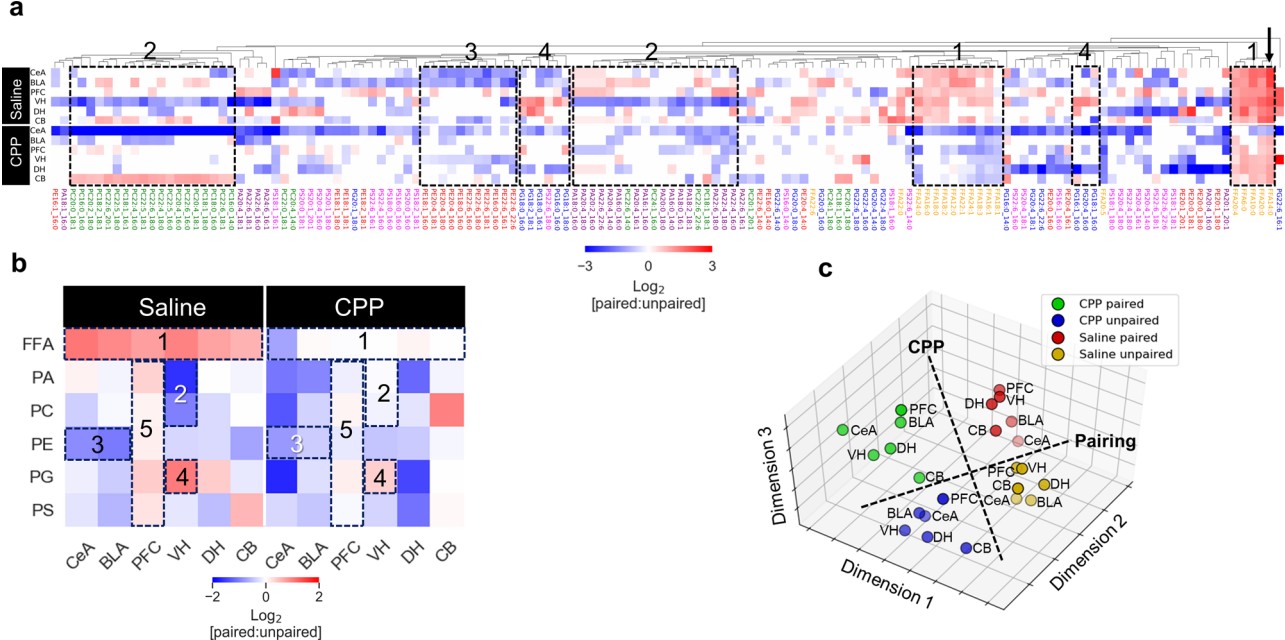

**Fig. 3 FFA and phospholipid responses to AFC. a** Hierarchical clustering heatmap, where each pixel represents the mean fold-change response (paired versus unpaired AFC) of the indicated analyte from 8 animals, in a given brain region. White pixels represent analytes whose change in abundance was not significant (two-tailed *t*-test *p* > 0.05). Dotted boxes highlight regions of the heatmap demonstrating strong activity-dependent change of a particular lipid class as described in the main text. The arrow shows the particularly strong response of myristic acid (C14:0). **b** Simplified heatmap indicating total response (paired versus unpaired AFC) of all lipids of a given class in each brain region. Dotted boxes correspond to those in **a**. **c** Multivariate analysis of the 24 FFA profiles representing 6 brain regions and 4 AFC conditions (Supplementary Fig. 4), where each dot represents 18 normalized mean FFA concentrations observed across 8 animals. The isomap algorithm was used for non-linear dimensionality reduction from 18 to 3 dimensions. The 3D projection has been manually rotated to highlight the differences between the profiles obtained for each AFC experimental condition. Dashed lines represent a manual assignment of the axes around which the profile datapoints were resolved, according to CPP treatment and AFC pairing. CeA—central amygdala, BLA—basolateral amygdala, PFC—prefrontal cortex, VH—ventral hippocampus, DH— dorsal hippocampus, CB—cerebellum, PA—phosphatidic acid, PC—phosphatidylcholine, PE—phosphatidylethanolamine, PG—phosphatidylglycerol, PS—phosphatidylserine. Source data are provided as a Source Data file.

in response to AFC. The AFC experimental framework used in this study allowed us to determine whether these brain region-specific changes to the FFA landscape correlated with a lasting learned response to fear conditioning. Immediately following AFC (Fig. 2b, 0 h) and for several hours thereafter, saline- and CPP-treated animals typically showed a very similar freezing response to a tone-only stimulus, indicative of acquisition of an identical short-term learned response to paired stimuli, in an NMDA receptor-independent manner. However, subsequent consolidation of fear memory is NMDA receptor-dependent[45], as it was blocked by CPP. 24 h after AFC (Fig. 2b, 24 h), saline-treated animals maintained the freezing response to tone, in contrast to CPP-treated animals which displayed a significantly attenuated response. Lipids were assayed 2 h post-AFC, at which time point the FFA response to paired AFC stimuli was consistently different between saline- and CPP-treated animals, even though both groups exhibited similar freezing behaviour at this stage. These data suggest that changes to the FFA landscape across the brain elicited by the short-term response to AFC are required for the consolidation of long-term memory. Blocking these profile changes through inhibition of NMDA receptors correlates with a failure of memory consolidation.

To confirm the relationship between quantitative changes in the FFA response and memory acquisition, we assessed the relationships between the FFA profiles obtained for each of the 24 samples (6 brain regions for 4 experimental conditions) using multivariate analysis. The 3456 individual FFA measurements were condensed into an 18 FFA x 24 sample array where each array element represented the mean measurement of an FFA

across 8 rats. The array was analyzed by non-linear dimensionality reduction analysis (NLDR) using the isomap algorithm, treating the array as 24 samples of 18 dimensions and mapping the data to a lower dimensional space where each successive dimension explained progressively smaller amounts of variation in the data[47]. By clustering the profiles based on their similarity in higher dimensional space, NLDR analysis confirmed the broad FFA responses to CPP treatment and AFC stimuli. NLDR revealed that profiles could be readily grouped according to the four experimental conditions (Fig. 3c), which correlated with long-term memory consolidation (Fig. 2b). Interestingly, similar clustering was not observed for any of the phospholipid classes (Supplementary Fig. 3c), likely reflecting the more complex response of the component species across different brain regions.

**FFA responses to AFC are strongest in the amygdala.** In terms of both the absolute change and fold change in abundance (Supplementary Fig. 3a, b), the FFA responses to AFC in saline- and CPP-treated animals were most strikingly demonstrated in the amygdala, particularly the central amygdala (Fig. 4a, b, Supplementary Fig. 4). These changes were dominated by saturated FFAs (Fig. 4b), particularly myristic acid, and to a lesser degree palmitic acid. Arachidonic acid was the most strongly responding unsaturated FFA (Fig. 4a). For all FFAs, CPP treatment dramatically reduced the response to paired AFC. Although the absolute responses of FFAs to fear memory acquisition were lower in the dorsal and ventral hippocampus, the response of arachidonic acid relative to other FFAs was elevated in these regions

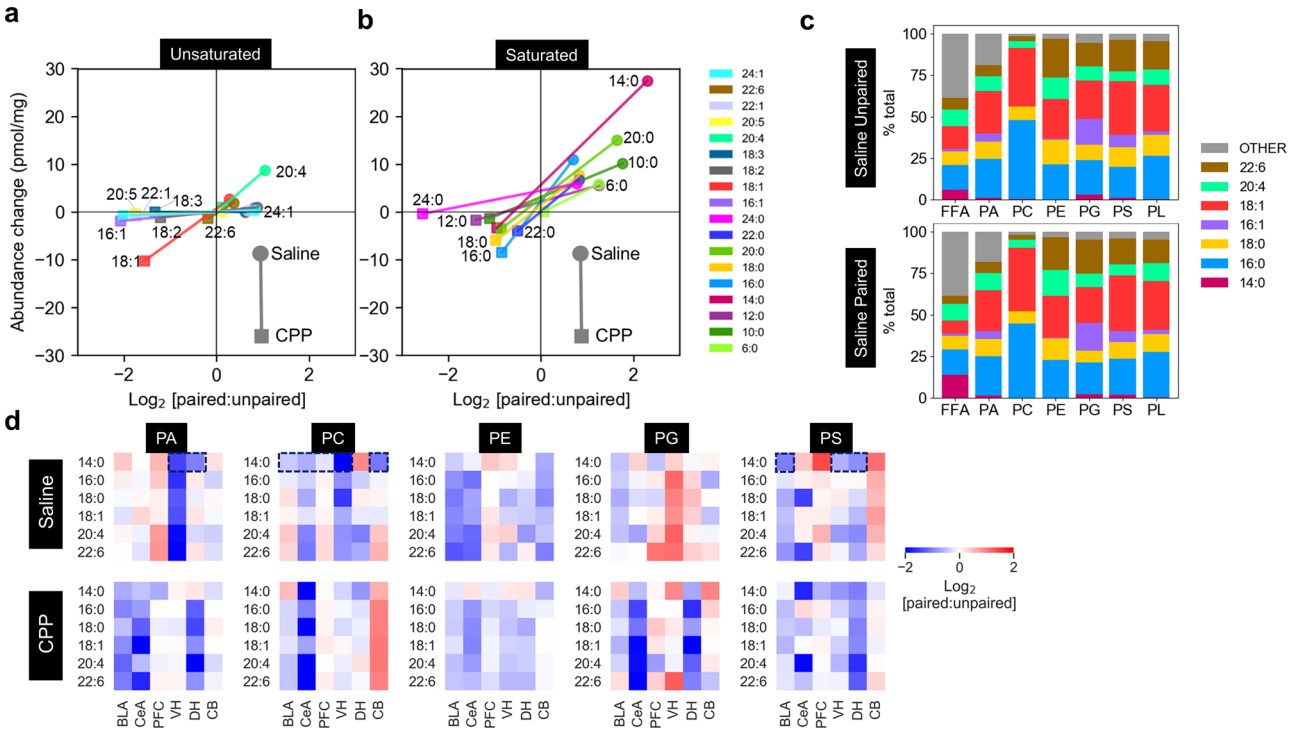

**Fig. 4 Myristic acid (C14:0) dominates the FFA response to AFC. a, b** Unsaturated and saturated FFAs in the central amygdala. Scatterplots show the AFC-induced absolute ($y$ axis) and $\log_2$ fold change ($x$ axis) in mean abundance for each FFA in saline- (round) and CPP- (square) treated animals. For all FFAs, CPP treatment resulted in a decreased response to paired AFC. **c** Most common acyl chain distribution across FFA and phospholipid class. Each coloured bar represents the summed abundance of all lipids containing a given acyl chain as a percentage of the summed abundance of all acyls for each lipid class. **d** Response of phospholipids containing the most common acyl chains. Each pixel in the heatmap represents the AFC-induced change in summed abundance of those species in each class containing a given acyl chain. Dotted boxes indicate brain regions where AFC induces a decrease in myristoyl containing phospholipids that could potentially represent substrates for phospholipase A mediated release of myristic acid. CeA—central amygdala, BLA— basolateral amygdala, PFC—prefrontal cortex, VH—ventral hippocampus, DH—dorsal hippocampus, CB—cerebellum, PA—phosphatidic acid, PC—phosphatidylcholine, PE—phosphatidylethanolamine, PG—phosphatidylglycerol, PS—phosphatidylserine. Source data are provided as a Source Data file.

(Supplementary Fig. 4). It is worth noting that the 6–8 week animals used in this study represent mid-late adolescent males[48–51]. Heightened emotional responses to negative events during adolescence are well documented, and adolescent rats have been demonstrated to show dramatically reduced extinction of fear-induced freezing behaviour compared to adults[52]. The heightened FFA response in the amygdala observed in this study is likely to at least partly reflect this developmental process.

While it is reasonable to speculate that increases in FFAs correlate with decreases in phospholipids due (at least in part) to phospholipase activity in the synapse during AFC, these substrate–product relationships are not immediately evident because phospholipids are present in brain tissue at far higher concentrations than FFAs, and a given FFA species can potentially be derived from the enzymatic processing of multiple phospholipids and other substrates such as lysophospholipids and acylglycerols. When the contribution of the most common acyl species across the measured phospholipid classes was summed (Fig. 4c), the distribution of FFA acyl species did not simply mirror the distribution of acyl species in any particular phospholipid class, nor of total phospholipids. This suggests that some degree of phospholipase specificity occurs during memory acquisition, as particularly evidenced by the high relative abundance of free myristic acid compared to the low relative abundance of myristoyl-containing phospholipids (Fig. 4c). Of note, in the central amygdala, PA/PC/PS phospholipids containing 20:4_14:0; 22:6_14:0 and 16:0_14:0 decreased (Fig. 3a). Conversely, free palmitic acid does not mirror its high relative

abundance in phospholipids, particularly phosphatidylcholine. Our data also suggests a strong degree of brain region specificity in phospholipid substrate processing. The strong activity-dependent increase in myristic acid across the brain was not mirrored by complementary decreases of myristoyl-containing phospholipid substrates in all of these regions (Fig. 4d).

## Discussion

To the best of our knowledge, this study represents the first comprehensive survey of the changes in the phospholipid metabolite landscape in response to memory acquisition. Using our lipidomics approach in conjunction with AFC, we have demonstrated that (i) the distribution of phospholipids and FFAs is heterogeneous across the rat brain, with the highest concentrations being found in the amygdala, (ii) phospholipid and FFA profiles change across the brain in response to AFC in an activity-dependent manner, (iii) these changes are characterized by increases in saturated FFAs (particularly myristic acid) and to a lesser degree unsaturated FFAs (particularly arachidonic acid) potentially released from phospholipid substrates, and (iv) blocking these changes with the NMDA receptor antagonist CPP correlates with a failure of long-term memory consolidation but has no effect on short-term responses to AFC. Together our results suggest that FFA changes regulated by NMDA receptors are required for learning and memory formation.

NMDA receptors that trigger LTP are essential for memory consolidation[53–57], and we have now shown that this is preceded

by significant changes in the phospholipid metabolite landscape in vivo. We have previously demonstrated that neural stimulation in vitro results in similar changes[39] suggesting that saturated FFAs, in addition to unsaturated arachidonic acid, play significant structural and signalling roles in neurotransmission and synaptic plasticity. Although the mechanisms by which these FFAs contribute to memory consolidation are still to be determined, a considerable body of work has demonstrated the importance of co-translational protein myristoylation and post-translational protein palmitoylation, in synaptic plasticity and LTP, as well as retention of cognitive function during ageing[58]. Protein acylation broadly functions to drive increased protein–membrane and protein–protein interaction and dynamic targeting to specific subcellular locations, and is critical for the activity-dependent regulation of important protein pathways in the pre- and post-synapse[59–61]. Post-translational palmitoylation is mediated by the zinc finger DHHC-type (zDHHC) family of enzymes, which are essential for the regulation of neuronal morphology and synapse formation and function[20,21,62]. Palmitoylation/depalmitoylation of the post-synaptic scaffold protein PSD95, is crucial for synaptic plasticity as it promotes the interaction and organization of AMPA and NMDA receptors[63] (themselves targets for palmitoylation[64,65]) in the post-synaptic density. δ-Catenin palmitoylation mediates synaptic plasticity by coordinating changes in synaptic adhesion molecules such as cadherin[66]. Co-translational myristoylation of proteins by N-myristoyltransferase (NMT1 and NMT2) is similarly important. The calcium sensing proteins hippocalcin[67] and recoverin[68] use a $Ca^{2+}$/myristoyl "switch" to reversibly associate with the neuronal membrane in response to $Ca^{2+}$ influx. G-protein coupled receptor (GPCR) signalling is crucial to neuronal development and function, and different GPCR subunits may be myristoylated[69] in addition to their palmitoylation. G-proteins themselves may also be myristoylated: ADB ribosylation factor (ARF) GTPases, which contribute to vesicular trafficking and phospholipase activation among other functions[70], are myristoylated to modulate their membrane interactions. In light of the importance of protein acylation by myristic and palmitic acid, it is tempting to speculate that the increase in these saturated FFAs in response to AFC pairing functions to generate substrates for such post-translational modifications, which in turn modulate neuronal structure and function to contribute to learning and memory. As protein acylation occurs via acyl-CoA intermediates, more work is needed to assess the potential link between FFA generation and synaptic protein acylation during memory acquisition.

Quantitative brain lipidomics is a challenging endeavour complicated by the staggering number of lipid species, requiring a range of different analytical strategies, and the need to rapidly stabilize the post-mortem lipid environment. We addressed the first aspect through a targeted lipidomic strategy employing multiplex carboxy-terminal isotope labelling for FFAs and unique acyl MS fragmentation signatures for phospholipids. Post-mortem lipid stability was achieved using perfusion/rapid brain freezing and subsequent low-temperature sample preparation and lipid extraction protocols (other researchers have used focused microwave fixation[71] for this purpose). Our experimental approach allowed us to determine how these critical analytes changed across the rat brain in response to experimental conditions.

The detection of predominantly saturated FFAs across the brain suggests the potential importance of phospholipid processing by pathways other than the classical PLA₂-mediated release of unsaturated FFAs from the sn-2 position of canonical phospholipids[72]: (i) saturated FFAs may be released by PLA₁ from the sn-1 position of canonical phospholipids (saturated sn-1 acyl, unsaturated sn-2 acyl), also yielding unsaturated 2-acyl lysophospholipids. Importantly, a recent study on PLA₁ knockout

mice showed altered memory processing in these animals[73], in support of the importance of PLA₁ to neuronal function; (ii) saturated FFAs could also be released by PLA₂ from the sn-2 position of non-canonical phospholipids (saturated sn-2)[72]. In support of this, unambiguous non-canonical phospholipids such as PS22:6_22:6 (unsaturated sn-1/unsaturated sn-2) and PS18:0_16:0 (saturated sn-1/saturated sn-2) were indeed observed in this study, and it is likely that many (if not all) of the phospholipids defined herein in fact represented a mixture of canonical and non-canonical positional isomers (beyond the differentiation ability of our instrumentation).

Our study also points to brain region-specific FFA responses to AFC, with the largest absolute response being observed in the amygdala and prefrontal cortex. The involvement of the central and basolateral amygdala and the interaction of these nuclei with the dorsal and ventral hippocampus in various forms of fear conditioning has long been established[74–76], and lipid metabolite changes indicative of synaptic plasticity would be expected to be concentrated in these areas. In contrast, the primary involvement of the cerebellum in motor learning (although its contribution to non-motor processes has become increasingly apparent) makes it less likely to be a critical focus for synaptic plasticity in response to AFC[77,78]. The cerebellum is therefore a reasonable candidate for a "control" brain region against which to gauge the responses of other regions to AFC. Interestingly, the prefrontal cortex of the forebrain exhibited the second largest response to AFC, higher than that of the hippocampus. The interplay of the forebrain with the auditory cortex has been shown to be important for AFC[79]. The forebrain also modulates the stress response via the neural cell adhesion molecule (NCAM), the ablation of which in the forebrain impairs AFC in mice[80].

Previous studies have demonstrated that polyunsaturated FFA profiles do not have any influence on the passive electro-physiological properties of cortical neurons, in stark contrast with the change in paired pulse ratio, an in vitro measure of short-term memory, observed when these FFA profiles are modified in vivo[81]. These results belong to the small number of studies that have established a link between FFA profile and brain function[82]. Our study provides the first brain-wide profile of FFA changes with learning, and establishes the importance of further studies of the functional significance of phospholipids and their metabolites in memory formation. Our data demonstrate that specific saturated and unsaturated FFAs generated during the acquisition of fear memory correlate with the long-term consolidation of that memory. The mechanisms by which FFAs mediate this consolidation, whether through modulation of membrane properties, post-translational targeting of proteins to interact with membranes and other proteins, or via other lipid signalling pathways, remain unclear. Coupling new developments in lipidomics techniques[83–85] to animal learning and memory models will be invaluable in uncovering the specific role(s) of these FFAs in learning and memory.

## Methods
**Materials**. Saturated and unsaturated FFAs were obtained from Sigma-Aldrich. Phospholipid standards (PC 17:0/17:0, PE 17:0/17:0, PS 17:0/17:0, PA 17:0/17:0, PG 17:0/17:0, PC 12:0/13:0, PE 17:0/14:1, PS 17:0/14:1, PA 12:0/13:0 and PG 12:0/13:0) were purchased from Avanti Polar Lipids. 1,1-carbonydiimidiazole, 1,4 dioxane, triethylamine, iodomethane, iodomethane-$d_3$, iodoethane, formic acid, citric acid, disodium hydrogen phosphate ammonium formate, acetonitrile, 1-butanol, methanol and chloroform were purchased from Sigma Aldrich. All reagents were analytical grade or equivalent. All lipid extractions were performed in 2 mL polypropylene LoBind safe-lock tubes (Eppendorf).

**Ethics**. In all procedures, the care and experimental use of animals was in accordance with protocols approved by the University of Queensland Animal Ethics Committee (QBI/313/13/NHMRC).

**Auditory fear conditioning**. Male Sprague-Dawley rats (380 ± 15 g, 6–8 weeks of age) were obtained from the University of Queensland Biological Resources (UQBR) facility. Animals were housed under standard laboratory conditions with a 12-h light/dark cycle and food and water were available ad libitum. Experiments were conducted in sound attenuating boxes (35 ± 3 dB, Coulbourn Instruments). Four auditory fear conditioning (AFC) conditions were used: saline-treated, unpaired AFC stimuli; saline-treated, paired AFC stimuli; (6)-3- (2-carbox-ypiperazin-4-yl)propyl-1-phosphonic acid (CPP)-treated, unpaired AFC stimuli; and CPP-treated, paired AFC stimuli. Eight animals were used for each condition. Prior to conditioning the rats were injected intraperitoneally with either CPP in 0.9% sterile saline at a dose of 10 mg/kg, or sterile saline alone. For the paired groups, after 120 s of context habituation five auditory tones (5 KHz, 75 dB ± 3 dB, 20 s, inter-trial interval 60–90 s) were presented with a co-terminating footshock (0.6 mA; 0.5 s). The unpaired groups received the same number of tones and shocks but the two stimuli were never temporally associated (minimum delay 20 s). The animals were returned to their homecage for 2 h prior to tissue collection.

**Tissue collection**. Brain tissue was collected under conditions designed to minimize ischaemic and post-mortem lipid metabolism. Animals were deeply anaesthetized with isoflurane and transcardially perfused with ice-cold oxygenated artificial cerebrospinal fluid (ACSF) containing 88 mM NaCl, 2.5 mM KCl, 25 mM NaHCO₃, 10 mM D-glucose, 1.2 mM NaH₂PO₄, 1.3 mM MgCl₂ and 2.5 mM CaCl₂ to rapidly remove all blood and to chill the brain. The animals were quickly decapitated and their brains were extracted, snap-frozen in liquid nitrogen and stored at −80 °C. Each frozen brain was promptly transferred to the mounting block of a cryostat (NX70, Thermo Fisher Scientific) operating at −20 °C, and 80–100 µm thick sections were cut and transferred to glass slides within the cold cryostat. Slides were quickly transferred to a microscope (Olympus SZ51) next to the cryostat, and appropriate brain regions were rapidly dissected from each frozen slice and placed into 2 ml tubes (Eppendorf) on dry ice. The dissection of specific regions from a given slice took less than a minute. On average, 70 slices were cut for each brain. Each tube, representing a specific brain region, contained material dissected from approximately 20 sequential brain slices. All samples were stored at −80 °C until subsequent lipid extraction. Extraction procedures were also performed under cold conditions, and at no stage prior to this were brain samples allowed to thaw.

**Synthesis of FFAST isotopic-coded differential tags**. 3-hydroxymethyl-1-methylpyridium iodide (FFAST-124), 3-hydroxymethyl-1-methyl-d₃-pyridium iodide (FFAST-127) and hydroxymethyl-1-ethylpyridium iodide (FFAST-138) reagents were synthesized from commercially available CH₃I/CD₃I/C₂H₅I and 3-hydroxymethyl-pyridine. The synthesis of FFAST derivatives was performed in an analogous fashion to the procedure described previously[39,86]: In a glass tube, 100 mg of 3-hydroxy-methyl-pyridine were added to 200 mg of iodomethane (for FFAST-124), iodomethane-D₃ (FFAST-127) or iodoethane (FFAST-138), respectively. An additional 500 µl of 1,4 dioxane was added, and the solution was microwaved between 1–300 Win a CEM Discover microwave reactor heated to 90 °C for 90 min under gaseous N₂. The resulting solution was washed with 500 µl diethyl ether and then dried. The reaction was evaluated using thin layer chromatography, and the purity of all the derivatives was further determined to be >95% by ¹H nuclear magnetic resonance (NMR) spectroscopy.

**Free Fatty Acid (FFA) extraction and FFAST labelling**. Frozen brain tissue samples were homogenized in 0.5 mL of HCl (1 M) for 5 min using a tissue homogenizer in the cold room. 0.6 mL of ice-cold chloroform followed by 0.4 mL of ice-cold methanol: 12 N HCl (96:4 v/v, supplemented with 2 mM AlCl₃) were added to 200 µL of the homogenate. Once the mixture was completely vortexed, 0.2 mL of ice-cold water was added and centrifuged for 2 min at 12000 × g at 4 °C in a refrigerated microfuge (Eppendorf 5415 R). The upper phase was discarded and the tube containing the lower phase was dried in a vacuum concentrator (Genevac Ltd). The dried extracts were re-dissolved in 100 µL acetonitrile. The derivatization strategy was aimed at compounds with free carboxylic acid groups, and was performed according to a previously described procedure[39]. Briefly, 100 µL of FFA extracts in acetonitrile were mixed with 50 µL of 1,1-carbonyldiimidiazole (1 mg/mL in acetonitrile) and incubated for 2 min at room temperature. Subsequently, 50 µL of either FFAST-124 or FFAST-143 (50 mg/mL in acetonitrile, 5% triethylamine) was added. After mixing for 2 min, the mixtures were heated in a 50 °C water bath for 20 min. Finally, the 2 different isotopically labelled samples were mixed together and dried in a vacuum concentrator. They were then re-dissolved in 200 µL of an internal standard solution (2.5 µM in acetonitrile) that was prepared earlier by derivatization of the 18 FFA standards with the FFAST-138 label. The samples were then transferred into an auto-sampler vial for liquid chromatography mass spectrometry (LCMS) analysis. All samples were stored at −20 °C prior to analysis.

**Free fatty acid LCMS analysis**. LCMS analysis was performed on a 5500QTRAP tandem mass spectrometer with an ESI TurboIonSpray® source (AB Sciex), connected to a Shimadzu Nexera UHPLC equipped with a XB-C18 Kinetex column, 50 × 2.1 mm, 1.7 µm (Phenomenex). Instrument control and data acquisition were

performed using Analyst® 1.5.2 software. The instrument was operated in the positive ion mode under multiple reaction-monitoring conditions (Supplementary Table 1). The Turbospray temperature was set to 400 °C, the curtain gas flow to 30 psi, and the ion spray voltage to 4500 V. The collision energy, declustering potential and collision cell exit potentials were optimized and set to 55 V, 100 V and 13 V respectively. Liquid chromatography (LC) was performed at 400 µL/min and 65 °C using a gradient system consisting of solvent A (99% water with 0.1% formic acid (v/v)) and solvent B (90% acetonitrile with 0.1% formic acid). The column was preconditioned with 40% B for 30 min before analysis, and equilibrated for 5 min in 40% B prior to sample injection. Samples were reconstituted in 100 µL of mobile phase A, and 5 µL of sample was injected onto the column using a pre-treatment condition (45 µL water + 5 µL sample in 50 µL loop). Following sample injection the gradient conditions were: 40–100% B (3 min), 62% B (1 min), 100% B (1 min), and 40% B (1 min), for a total run time of 6 min. The first 0.5 min of the LC run was switched to waste to remove any excess underivatized FFAST tags. Data were acquired between 0.5 min and 6 min.

**Phospholipid extraction**. Frozen brain tissue samples from AFC rats were homogenized with five volumes of cold citrate buffer (30 mM) containing 1.25 ug/mL internal standard mixture (PC 17:0/17:0, PE 17:0/17:0, PS 17:0/17:0, PA 17:0/17:0 and PG 17:0/17:0) for 5 min using a tissue homogenizer in the cold room. Phospholipid extraction was performed essentially according to the method of Bligh and Dyer[87]. In brief, to 100 µL of the homogenate, 0.9 mL of ice-cold chloroform:methanol (1:2 v/v) was added and vortexed for one minute. To this, 0.3 mL of chloroform followed by 0.3 mL of 0.1 N HCl were added. The mixture was vortexed and incubated for one minute and then centrifuged for phase separation at 12000 × g in a refrigerated benchtop microfuge for 5 min at 4 °C. The lower organic phase containing phospholipids was collected and stored at −80 °C for further analysis.

**Phospholipid LCMS**. Analysis was performed on the Shimadzu Nexera UHPLC / 5500QTRAP tandem mass spectrometer system described above. Chromatographic separation was performed on a Kinetex 2.6 µm HILIC 100 Å column (150 × 2.1 mm) (Phenomenex, USA). The instrument was operated in negative ion mode under multiple reaction-monitoring conditions. The Turbospray temperature was set to 300 °C, the curtain gas flow to 30 psi and the ion spray voltage was −4500 V. The collision energy (CE) was set to −50 for all phospholipid species. The declustering potential (DP) and collision cell exit potential were optimized and were set to 100 and 13, respectively. The precursor masses of each phospholipid species in every class was selected as Q1 and the fatty acyl chains generated in negative ion mode were selected as Q3 respectively (Supplementary Table 1). HILIC chromatography elution conditions were set at 400 µl/min and 65 °C, using a gradient system consisting of solvent A (95% acetonitrile with 5% 50 mM ammonium formate, pH 3.55) and solvent B (50% acetonitrile, 45% water with 5% 50 mM ammonium formate, pH 3.55). After equilibration with 95% B, the gradient elution used was 0–40% B (1 min), 40–98% B (10 min), 98% B (2 min) and 98–40% B (1 min). The column was equilibrated at 95% B for 10 min before the next run. The injection volume of all the samples was 5 µl.

**Quantitative data processing and visualization**. Experimental FFA and phospholipid data were analyzed quantitatively with Multiquant® 3.03 (AB SCIEX), using the MQ4 peak picking / integration algorithm to manually determine the LCMS peak area for each analyte. For each FFA the abundance of each species (FFAST-124 or FFAST-127 labelled) was determined relative to the FFAST-138 labelled internal standard for that species, using a previously established calibration curve for that species. Triplicate calibration curves were generated using 6 serial dilutions of the FFAST-124 or FFAST-127 labelled FFA calibrators (0.05 ng/ml to 25 ng/ml in acetonitrile) combined with 2.5 ng/ml of the corresponding FFAST-138 labelled internal standard. A standard curve for each FFA was plotted as the ratio between analyte and internal standard peak area. Calibration slopes were calculated by linear regression without weighting ($R^2 > 0.99$). For experimental data, FFA quantification was performed using custom Python (python.org) scripting. The ratio of the analyte (FFAST-124 or FFAST-127) peak area to the internal standard (FFAST-138) peak area was divided by the calibration slope for that analyte to derive the concentration (ng/ml) of the analyte in the injected sample, which was converted to the molar concentration (pmol/µl) by dividing by the molecular weight of the labelled analyte. Tissue sample weight and extraction volumes were then used to normalize the concentration to pmol/mg tissue.

For phospholipids the abundance of the species within each phospholipid class was determined relative to a single standard for each class. Triplicate calibration curves were prepared for each of the phospholipid classes (PA, PC, PE, PG and PS). A 10 µg/mL standard mixture of PA 12:0/13:0, PC 12:0/13:0, PE 17:0/14:1, PG 12:0/13:0 and PS 17:0/14:1 was prepared in mobile phase A. These were serially diluted into 6 steps to prepare calibrators (0.15 µg, 0.312 µg, 0.625 µg, 1.25 µg, 2.5 µg and 5 µg/mL). The lower limit of detection (LLOD) and lower limit of quantitation (LLOQ) were up to 0.15 µg/mL and 0.60 µg/mL on column for all monitored phospholipids. To each calibrator, an internal standard mixture containing PC 17:0/17:0, PE 17:0/17:0, PS 17:0/17:0 and PA 17:0/17:0 was added to a final concentration of 1.25 µg/mL. A standard curve for each phospholipid class was

plotted as the ratio between analyte and internal standard peak area ratio of the same lipid class. Calibration slopes were calculated by linear regression without weighting ($R^2 > 0.99$). For experimental data, phospholipid quantification was performed using custom Python scripting essentially as described above for FFAs. For species with two distinct two distinct acyl chains the final abundance was calculated by combining the intensity information of the multiple reaction monitoring (MRM) peaks corresponding to each chain.

To visualize and assess the significance of changes in analyte concentration across brain regions and in response to experimental conditions, lipid data were further analyzed using custom-written Python 2.7.17 (python.org) scripts variously utilizing Matplotlib 2.2.5 (matplotlib.org), Pandas 0.24.2 (pandas.pydata.org), Numpy 1.16.6 (numpy.org), Seaborn 0.9.1 (seaborn.pydata.org), Scipy 1.2.3 (scipy.org) modules. For each analyte the eight measurements (one for each of the animals) were processed to remove outliers by median filtering and generate the mean and standard error of the mean (SEM). Stacked barplots of these data were generated using Pandas and Matplotlib. Heatmaps were generated using Pandas and Seaborn. For heatmaps the significance of the fold-change for each pixel was determined by Student's two-tailed $t$ test using scipy.stats.ttest_ind. Scatterplots were generated using Pandas and Matplotlib.

Animal freezing behaviour data was analyzed and visualized using Prism® 9.0 (Graphpad).

**Multivariate data analysis and visualization**. Qualitative multivariate analysis using dimensionality reduction to determine features and relationships within the total data for each lipid class was performed using custom Python scripting based on Python Scikit-learn 0.20.4 (sklearn, scikit-learn.org). Each datapoint for the analysis comprised the mean of the measurements of each of the individual species being measured for a given lipid class—e.g. 18 FFAs represent an 18-dimensional datapoint. 24 sample datapoints, consisting of the four experimental conditions for each of the six brain regions, were analyzed (e.g. basolateral-amygdala_saline_unpaired, basolateral-amygdala_saline_paired etc). Data were transformed by normalization, such that analytes summed to one for each datapoint. Non-linear dimensionality reduction (NLDR) to 3 dimensions used the isomap algorithm[47] implemented in sklearn.manifold.isomap, considering all 23 nearest neighbours in the higher dimensional manifold and using default settings otherwise. Dimensionally reduced data plotted in three dimensions allowed manual rotation around three axes to best visually determine the relationships between groups of datapoints, assisted by the ability to colour the datapoints based on whether they represented CPP/saline, or paired/unpaired results.

**Reporting summary**. Further information on research design is available in the Nature Research Reporting Summary linked to this article.

## Data availability

The data generated during this study, presented as quantified lipid abundances, may be downloaded from the publicly accessible University of Queensland Data Collection: https://doi.org/10.14264/12793dc. Source data are provided with this paper.

## Code availability

The Python scripts for quantitative and multivariate data analysis and visualization are available upon request. Requests for software should be addressed to f.meunier@uq.edu.au (F.A.M.)

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

## Acknowledgements

This work was supported by National Health and Medical Research Council (NHMRC) project grants GNT1128427 and GNT1120374, and Australian Research Council Grant CE140100007. F.A.M. is an NHMRC Senior Research Fellow (GNT1060075). We thank Rowan Tweedale for critical reading of this manuscript.

## Author contributions

F.A.M., V.K.N. and D.K. conceived the study. V.K.N. developed the FFAST protocol and performed the lipidomic data acquisition as part of a PhD program under the supervision of D.K. and F.A.M., B.G.V. implemented the phospholipid LCMS protocol and acquired lipidomic data as part of a PhD program under the supervision of T.P.W. and F.A.M., T.P.W. performed data analyses, and wrote the manuscript with F.A.M., with input from B.G.V. and the other authors. Fear memory and pharmacological manipulation experiments were designed and implemented by F.W. and brain tissue excised by R.K.S., in the laboratory of P.S., A.H. visualized quantitative and multivariate analyses.

## Competing interests

The authors declare no competing interests.
