## [Peer Review File · Nature Communications]

Reviewers' comments:

Reviewer #1 (Remarks to the Author):

In their manuscript, authors assess concentrations of free fatty acids in several brain regions of rats exposed to fear conditioning, as well as controls. Using this classical paradigm, authors show substantial alterations in the concentration of free fatty acids (FFAs) after the stimulation, particular to specific regions. These observations are novel and indicate that lipidome components of the brain are much more active participants in the brain functionality, as previously thought.

While the experimental part of the work appears solid, results representation might be improved and made more informative. The results are novel and are likely to attract attention from readers from across the field, so I would strongly recommend to the authors explore more than one way to represent their results to make them more readily understandable.

For instance, authors use the same type of bar plots to display differences in FFA abundance in each brain region after certain treatment on Fig 2 and 3, and a somewhat different design for Fig 4 and 5. Plus there is also the same information is given for saturated/unsaturated FFAs, as well as individual FFAs in the supplement, for each condition. It is almost impossible to grasp the full picture of reported FFA changes from these illustrations. I think it would be useful to show the summary dynamic of each FFA in each of brain region, including the statistical analysis, on united summary plots. This does not need to be a large figure or table, as the information could simply be represented, for instance, as a heat map with colors reflecting absolute or relative concentrations and differences. The significant differences could be highlighted with colored bordering or by other means. This would also help to support the authors' statements about particular FFAs. Now, based on the summary box plots, it is not apparent, and significance information is hidden in the supplement.

Furthermore, a global analysis of FFAs changes across conditions, such as the one underlying the plots suggested above, might allow authors to cluster FFAs in a way which explains sample separation shown in Fig 6d. There should the FFAs with difference profile underlying this separation and it would be informative to see at least the trends.

The paper would also benefit from a summary schematics of all treatment and their relationship. Now, authors do a good job to provide a simple schematics for each experiment, but it does not help to understand how these different treatments are linked in the study paradigm. While this is described in the text, one needs to be prepared that some of the readers might not be readily familiar with this type of experiments, so a lot of things which are "self-explanatory" to the authors, might not be such for these readers.

The same is true for FFAST method. It would help if it were described at the lay audience level in the introduction, not just mentioned.

Reviewer #3 (Remarks to the Author):

Wallis et al. is an exploration of an understudied and potentially highly innovative area of free fatty acids (FFAs) in the brain, and their potential function. Strengths of the study include using isotope labeling multiplex analysis of FFAs (FFAST, a method previously published by the corresponding author) to map the expression of many FFAs following behavioral training, and providing the link to

Brain Fat MAP resource. However, the functional links between the FFA distribution findings and behavioral data of the study is tenuous at best and the data are too preliminary. Sections involving NMDA antagonist CPP are abrupt and not well rationalized. There are also objective errors in the methodology and statistical analyses inflating type 1 error that inevitably led to false or overstated claims. This study may be suitable for publication in this journal with additional data with clear functional outcomes and a complete re-analysis, see below.

Major issues:

1. Observing the figures, and in particular the extended data Figures 3-8, it is clear that all the FFAST data are derived from a single experiment with 5 groups (tone-only (also referred to as baseline) vs saline-unpaired vs saline-paired vs CPP-unpaired vs CPP-paired). For example, saline-unpaired group data (SU) in Figure 2b is identical to saline-unpaired group data (SU) in Figure 3b. That is, each figure 'cherry-picked' two groups to compare from the five-group single experiment. Such repeated t-tests drastically inflate type 1 error rate, which may explain why some seemingly subtle effects (e.g., Figure 4b in cerebellum CB and forebrain FB) are still reported to be significant. The correct analysis is as follows: a one-way ANOVA with all 5 groups, with Tukey's multiple comparisons to control for type 1 error rate. This way, statements can be made about significant changes following different behavioral protocol. This corrected analysis can only be done if the 5 groups were run concurrently. Also, this would significantly improve the presentation of data – instead of cherry-picking 1-2 groups at a time per figure and analyses, each graph can show one brain region but all five groups, with type 1 error controlled statistical analyses. This way the behavioral manipulation can be explicitly compared to 'baseline' and unpaired control groups.
2. A true baseline control group is missing. The 'baseline' group reported in Figure 1 is not a true baseline with the animals being exposed to an environment with a tone. This is important because the stated rationale for the study to select auditory fear conditioning as the behavioral treatment is due to Figure 1 showing significantly high 'baseline' FFA levels in the basolateral and central amygdala. However, these levels are not resting baseline as authors claim. The tone-only exposure in fact would bias the selection of the amygdala, with amygdala receiving direct and indirect auditory input from the thalamus. Therefore, a true naive group is necessary to actually discern whether the distribution of FFAs is indeed concentrated in the amygdala without any auditory stimulation. To be really rigorous, a shock-only control group is also necessary to discern that tone and shock presentations don't have summation effects on FFA expression.
3. Figures suggest that saline-unpaired and CPP-unpaired received a different unpaired protocol (see Figures 3 & 5). Considering that unpaired protocol itself causes drastic increases in FFA expression compared to tone-only group, any subtle differences in unpaired protocol can explain the differences between saline-unpaired and CPP-unpaired data. Please clarify whether the rats received random/mixed unpaired, or a set unpaired protocol but the same across saline vs CPP conditions. If not, then the saline vs CPP comparisons are uninterpretable.
4. Behavioral test data are missing unpaired groups to show that indeed associative learning is selectively shown in paired groups.
5. Considering that most rats receive saline or CPP injections before conditioning, it is puzzling that conditioning data are not presented to confirm that CPP had no effects on freezing during conditioning. Conditioning data should be presented, as it may correlate with FFA expression.
6. Related to the previous point, all of FFA expression may be related to freezing during conditioning and is not reflecting 'consolidation'. The evidence for the involvement of differential FFA expression in consolidation of conditioned fear is very weak – just because there are a few fatty acids in each behavioral condition that are regulated differently is not evidence that those changes are related to consolidation mechanisms. For this publication to be acceptable for this journal, the authors need to do an experiment involving injection of FFA actually changes behavior (e.g., Moon et al. 2014 *Metabolism* 1131-1140).

7. All brains were collected 2 hours following behavioral manipulation – there is no evidence that this is the point of ‘consolidation’ that CPP is targeting. A time-course of FFA expression, at least in the saline-paired group, is necessary to show correlational evidence of consolidation.
8. The rationale for investigating the effect of CPP injection is very weak. Is there a molecular pathway that could be linked up in the context of NMDA signaling?
9. Methods should state procedures for the behavioral treatment given to ‘tone-only’ group, as well as unpaired group. Methods should be re-written to clearly reflect that their ‘baseline’ was tone-only.
0. N is unclear – it is implied as n=8 per group, but Figure 1 legend reports 4 animals, and examining the individual data points in extended data Figures 3-8 show less than 8 per group, and changing n value across different FFA. In Figure 1, it is implied that 4 animals are pooled, authors should clarify that the data generated from 4 samples are tracked individually.
1. Have a single timeline showing all 5 FFA groups, with a clear indication of how long before conditioning saline/ CPP was injected, and when the brains were microdissected (e.g. 2 hours).
2. Revise strong claims and speculations – such as NMDA-receptor activity controls the generation of these FFAs (page 9), can be re-written as ‘associated’ not controlled (if the statistics hold up), and the effect does not seem limited to associative learning but stimulation (CPP also changes unpaired groups). Mention of beta-oxidation is also out of the blue (page 5). Instead, the various changes in different brain regions should be discussed with a circuitry perspective.

Minor issues:

1. Change the label of forebrain ‘FB’ to prefrontal cortex (PFC) – it is clear that the FB region was quite specific to the medial prefrontal cortex.
2. Figure legend for each color plot should also state the name of the FFA (e.g., Palmitic (C 16:0)). It is difficult to understand the stacked color bar graphs based on the legend’s use of the numeric shorthand nomenclature (e.g. 16:0) for the FFAs.
3. In text, only use the names of acid. Sometimes both the name and number are used, and sometimes just a number is used. With the revised legend (comment above), it should be easy to understand if only the names are used in text.
4. For the highlighted FFAs in text, state the corresponding % change – it has been done for some, but not all.
5. Abbreviations in general are inconsistent, some undefined, and some defined but used in a full form. A thorough proof-reading is necessary.
6. References to the corresponding author’s cell culture study (Narayana et al., 2015) would benefit from specifying the species of cells (rats as in this study).
7. Fig 1. missing bregma AP location for the cerebellum
8. Panel c of a few figures are wrongly introduced, as paired comparison, when no paired groups are present in some figures.
9. Replace the misleading word ‘Absolute response’ to ‘difference score’. When correctly presenting data with all 5 groups (6-7 including the requested controls), difference score can be calculated from the baseline group.

Reviewer #4 (Remarks to the Author):

There are well established methods to fix the brain in vivo (See review Murphey EJ 2010) that have been established for over 30 years to measure brain FFA. Unfortunately, the ischemia with a 10 second delay can dramatically alter brain FFA, let alone 10 minutes. However, even considering that, the FFA values presented in this study are higher than what is often reported for the total pool. In short the numbers are about 10000 times off what is accepted by the field.

Reviewers' comments:

Reviewer #1 (Remarks to the Author):

In their manuscript, authors assess concentrations of free fatty acids in several brain regions of rats exposed to fear conditioning, as well as controls. Using this classical paradigm, authors show substantial alterations in the concentration of free fatty acids (FFAs) after the stimulation, particular to specific regions. These observations are novel and indicate that lipidome components of the brain are much more active participants in the brain functionality, as previously thought.

1- While the experimental part of the work appears solid, results representation might be improved and made more informative. The results are novel and are likely to attract attention from readers from across the field, so I would strongly recommend to the authors explore more than one way to represent their results to make them more readily understandable.

We thank the reviewer for these encouraging remarks and have completely changed the data presentation to address this comment, as detailed below.

2- For instance, authors use the same type of bar plots to display differences in FFA abundance in each brain region after certain treatment on Fig 2 and 3, and a somewhat different design for Fig 4 and 5. Plus there is also the same information is given for saturated/unsaturated FFAs, as well as individual FFAs in the supplement, for each condition. It is almost impossible to grasp the full picture of reported FFA changes from these illustrations. I think it would be useful to show the summary dynamic of each FFA in each of brain region, including the statistical analysis, on united summary plots. This does not need to be a large figure or table, as the information could simply be represented, for instance, as a heat map with colors reflecting absolute or relative concentrations and differences. The significant differences could be highlighted with colored bordering or by other means. This would also help to support the authors' statements about particular FFAs. Now, based on the summary box plots, it is not apparent, and significance information is hidden in the supplement.

We are highly appreciative of these detailed suggestions, and have now presented our central findings as a series of ordered heatmaps, volcano plots and scatterplots which allow the reader to assess at a glance the abundance of FFAs across the brain, and the changes in abundance following paired auditory fear conditioning (AFC). Using this presentation, the FFAs which responded most strongly to paired AFC can be readily identified, and the effect of CPP on these responses is easy to ascertain.

3- Furthermore, a global analysis of FFAs changes across conditions, such as the one underlying the plots suggested above, might allow authors to cluster FFAs in a way which explains sample separation shown in Fig 6d. There should the FFAs with difference profile underlying this separation and it would be informative to see at least the trends. The new heatmap presentation of the data now clearly shows the FFAs which contribute to the differences in profiles correlating with the NLDR analysis. We thank the reviewer for prompting this approach in visualizing our data set.

4- The paper would also benefit from a summary schematic of all treatment and their relationship. Now, authors do a good job to provide a simple schematic for each experiment, but it does not help to understand how these different treatments are linked in the study paradigm. While this is described in the text, one needs to be prepared that some of the readers might not be readily familiar with this type of experiments, so a lot of things which are "self-explanatory" to the authors, might not be such for these readers.

In light of our move away from bar plot-based presentation with associated experimental schematics, we have consolidated the AFC experimental design and behavioural responses into Fig. 2a and 2b, which logically flows on to the results in Fig. 2c and 2d.

5- The same is true for FFAST method. It would help if it were described at the lay audience level in the introduction, not just mentioned.

We now visually summarised the FFAST process in Figure 1c, and the accompanying figure legend now explains the process in detail. This figure now encompasses the entire phospholipid analysis pipeline as well as the FFAST.

Reviewer #3 (Remarks to the Author):

Wallis et al. is an exploration of an understudied and potentially highly innovative area of free fatty acids (FFAs) in the brain, and their potential function. Strengths of the study include using isotope labelling multiplex analysis of FFAs (FFAST, a method previously published by the corresponding author) to map the expression of many FFAs following behavioral training, and providing the link to Brain Fat MAP resource. However, the functional links between the FFA distribution findings and behavioral data of the study is tenuous at best and the data are too preliminary. Sections involving NMDA antagonist CPP are abrupt and not well rationalized. There are also objective errors in the methodology and statistical analyses inflating type 1 error that inevitably led to false or overstated claims. This study may be suitable for publication in this journal with additional data with clear functional outcomes and a complete re-analysis, see below.

To address the points raised by the reviewer, we have now carried out additional work including:

- 1- A complete re-analysis of our dataset and more appropriate representation of the results as heat map and volcano plots allowing easier evaluation of the changes observed.
- 2- A better justification of our behavioral analysis and its correlation with the observed changes.
- 3- A new study to unravel the entire phospholipid response to memory acquisition thereby providing tantalizing evidence for substrate specificity leading to the FFAs response to memory acquisition. To carry out this study we have used samples from fear conditioned animals and have surveyed the memory-dependent change in the phospholipid landscape, using ESI-MS/MS to quantify the response of the most abundant 135 phospholipid species across 5 classes. We have demonstrated a very complex activity-dependent decrease in phospholipid abundance, varying by lipid class and brain region, which is heavily modulated by CPP. We consider that these findings strengthen the link between activity dependent phospholipase activity driving increases in FFAs which correlate with consolidation of long-term memory.
- 4- With regards to our use of NMDA antagonist CPP - Our initial behavioural analysis was carried out to survey of the response of the rat brain FFA “landscape” to memory acquisition, using auditory fear conditioning (AFC) as a well-established memory model. Rats subjected to paired AFC stimuli retain a long-lasting learned “freezing” response. Rats treated with CPP prior to AFC fail to retain the learned response. However, both control (saline) and CPP animals exhibit an identical “freezing” response immediately following AFC. Analysing FFAs in the short term following AFC allowed us to determine how FFAs respond to AFC, and how this response is modified in the CPP-treated animals. We hope these statements clarify our aim in this study.

Major issues:

1. Observing the figures, and in particular the extended data Figures 3-8, it is clear that all the FFAST data are derived from a single experiment with 5 groups (tone-only (also referred to as baseline) vs saline-unpaired vs saline-paired vs CPP-unpaired vs CPP-paired). For example, saline-unpaired group data (SU) in Figure 2b is identical to saline-unpaired group data (SU) in Figure 3b. That is, each figure ‘cherry-picked’ two groups to compare from the five-group single experiment. Such repeated t-tests drastically inflate type 1 error rate, which may explain why some seemingly subtle effects (e.g., Figure 4b in cerebellum CB and forebrain FB) are still reported to be significant. The correct analyses is as follows: a one-way ANOVA with all 5 groups, with Tukey’s multiple comparisons to control for type 1 error rate. This way, statements can be made about significant changes following different behavioral protocol. This corrected analyses can only be done if the 5 groups were run concurrently. Also, this would significantly improve the presentation of data – instead of cherry-picking 1-2 groups at a time per figure and analyses, each graph can show one brain region but all five groups, with type 1 error controlled statistical analyses. This way the behavioral manipulation can be explicitly compared to ‘baseline’ and unpaired control groups.

As detailed in response to reviewer 1, point 2, we have modified our data presentation to largely heatmap and scatterplot-based figures which allow the reader to readily assess the changes to the FFA landscape activated by paired AFC, and how these changes are modified by CPP. The significance of the change of each analyte is built into this presentation, such that statistically insignificant abundance changes (t -test $p > 0.05$) are represented by blank pixels. In the remaining figures containing bar plots, we have applied *post hoc* p value correction using the Holm-Sidak algorithm in consideration of potential type 1 errors.

2. A true baseline control group is missing. The ‘baseline’ group reported in Figure 1 is not a true baseline with the animals being exposed to an environment with a tone. This is important because the stated rationale for the study to select auditory fear conditioning as the behavioral treatment is due to Figure 1 showing significantly high ‘baseline’ FFA levels in the basolateral and central amygdala. However, these levels are not resting baseline as authors claim. The tone-only exposure in fact would bias the selection of the amygdala, with amygdala receiving direct and indirect auditory input from the thalamus. Therefore, a true naive group is necessary to actually discern whether the distribution of FFAs is indeed concentrated in the amygdala without any auditory stimulation. To be really rigorous, a shock-only control group is also necessary to discern that tone and shock presentations don’t have summation effects on FFA expression.

Whereas we initially used “tone-only” animals as our control group, we now use saline unpaired AFC animals. Use of tone-only animals allowed us to determine how the stress of footshock affected FFA levels. Use of context only animals which are placed in the AFC apparatus but subjected to neither tone nor shock would have allowed us to determine how exposure to the AFC environment affected FFA levels. A truly naïve cohort of animals that had never been exposed to the AFC apparatus would have allowed us to determine how the entire AFC process itself affected FFA levels. In the context of this study, where we are trying to determine how memory acquisition (paired AFC) affects FFA levels, unpaired animals that are treated in an otherwise identical manner to paired animals represents the most appropriate control.

3. Figures suggest that saline-unpaired and CPP-unpaired received a different unpaired protocol (see Figures 3 & 5). Considering that unpaired protocol itself causes drastic increases in FFA expression compared to tone-only group, any subtle differences in unpaired protocol

can explain the differences between saline-unpaired and CPP-unpaired data. Please clarify whether the rats received random/mixed unpaired, or a set unpaired protocol but the same across saline vs CPP conditions. If not, then the saline vs CPP comparisons are uninterpretable. The saline- and CPP-treated animals received identical tone/shock patterns. This has now been clarified in Fig. 2. We apologise for any confusion in our presentation.

4. Behavioral test data are missing unpaired groups to show that indeed associative learning is selectively shown in paired groups.

Long-term behavioural data could obviously not be obtained from the cohort of animals sacrificed subsequent to AFC for FFA analysis. We have therefore presented a typical dataset obtained at 24 h from an independent AFC experiment. The AFC paradigm is widely used in learning and memory research and behavioural responses are well-established. The critical point of our experiment is that the formation of associative memory which is engaged by the conditioned (auditory) stimulus – unconditioned (electrical) stimulus (CS-US) pairing but not in the unpaired group^{1,2} is significant in the context of this study, and is therefore presented. We have amended Fig. 2b to include data from the unpaired group immediately following AFC.

5. Considering that most rats receive saline or CPP injections before conditioning, it is puzzling that conditioning data are not presented to confirm that CPP had no effects on freezing during conditioning. Conditioning data should be presented, as it may correlate with FFA expression. During the conditioning procedure animals receive tones and aversive footshocks. Footshock itself induces a range of behaviours that are defensive and can also potentially evoke responses to nociceptive inputs activated by the shock. Hence, scoring freezing, which is a specific form of cessation of movement³, during acquisition is a poor marker of pairing-induced synaptic enhancement (see also⁴). We have now included conditioning data in Extended Data Fig. 1, to show that CPP has no effect on acquisition of the freezing response over the course of AFC.

6. Related to the previous point, all of FFA expression may be related to freezing during conditioning and is not reflecting ‘consolidation’. The evidence for the involvement of differential FFA expression in consolidation of conditioned fear is very weak – just because there are a few fatty acids in each behavioral condition that are regulated differently is not evidence that those changes are related to consolidation mechanisms. For this publication to be acceptable for this journal, the authors need to do an experiment involving injection of FFA actually changes behavior (e.g., Moon et al. 2014 *Metabolism* 1131-1140).

Animals were fear conditioned with neutral tones contingently paired with footshocks. This procedure led to them learning that the tone predicts an aversive outcome, after which the animals then showed a defensive response when the tone was presented (freezing). As can be seen in Fig 2b, following conditioning, animals tested immediately after the learning session (0 hours) freeze to the tone and following consolidation (24 hours later) also freeze. CPP blocks NMDA receptors and hence long-term potentiation (LTP)⁵, and LTP is critical for memory consolidation. As such, CPP-treated animals, when tested immediately following the training session (0 hours) freeze to the tone, but not when tested 24 hours later.

Lipids were assayed 2 hours after the pairing session, partly to allow the animals to calm so that we would not just be measuring stress based increases in FFAs. We show that in these 2 hours following conditioning the FFA landscape changes, and argue that these changes are initiated by the induction of LTP in the amygdala. In animals that received CPP, and froze immediately after the training session, the changes in FFA are not seen consistent with our suggestion that these changes are reflecting the induction of LTP. The fact that CPP treated

animals freeze immediately after the learning but have an altered FFA response clearly shows that the changes in FFAs are not the result of animals freezing.

One of our hypotheses for the increase in FFAs described in our manuscript is that membrane phospholipids are being used as substrates for a number of phospholipases to generate lyso-phospholipids and FFAs. We have explored this hypothesis *in vivo* and have now included our data on the entire phospholipid landscape change in response to memory acquisition. As expected, the levels of phosphatidylcholine are reduced suggesting that they are used as substrate to generate this range of FFAs through the action of phospholipases.

Injecting FFA as carried in Moon et al, 2014, would not be appropriate in our view because, it cannot rescue the loss of phospholipid substrate nor the parallel increase in lyso-phospholipids which are also likely to also have an effect⁶.

[redacted]

7. All brains were collected 2 hours following behavioral manipulation – there is no evidence that this is the point of ‘consolidation’ that CPP is targeting. A time-course of FFA expression, at least in the saline-paired group, is necessary to show correlational evidence of consolidation.

In this paper, we propose that changes in FFAs reflect biochemical changes associated with LTP which are required for memory consolidation. In addition to reducing the stress-based FFA response, we chose the 2 hour point for this analysis as most early biochemical changes that are involved in synaptic plasticity (eg cFOS) peak at this time. Testing animals the next day would introduce a large number of confounding factors relating to their activity including feeding and sleeping etc that might also impact FFA levels. The fact that CPP blocks these changes supports our proposal that they are reflecting generation of LTP and the process of consolidation.

8. The rationale for investigating the effect of CPP injection is very weak. Is there a molecular pathway that could be linked up in the context of NMDA signaling?

NMDA receptors that trigger LTP are essential in memory consolidation - something which has been repeatedly demonstrated in multiple models⁹⁻¹³. CPP is a selective blocker of NMDA receptors that easily crosses the blood brain barrier and thus can be delivered intraperitoneally. There are other avenues to disrupt LTP, however most of these are pharmacological compounds that do not cross the blood brain barrier (eg. APV¹⁴). These compounds require the insertion of cannula into the brain which might impact FFAs level through inflammation. We have now included these references in the appropriate section.

9. Methods should state procedures for the behavioral treatment given to ‘tone-only’ group, as well as unpaired group. Methods should be re-written to clearly reflect that their ‘baseline’ was tone-only.

We have re-written our Methods to reflect that fact that we no longer use “tone-only” animals in order to simplify the message.

0. N is unclear – it is implied as n=8 per group, but Figure 1 legend reports 4 animals, and examining the individual data points in extended data Figures 3-8 show less than 8 per group, and changing n value across different FFA. In Figure 1, it is implied that 4 animals are pooled, authors should clarify that the data generated from 4 samples are tracked individually.

8 animals were used for each experimental condition. For each analyte, outlier reduction (median filtering) was performed, which removed between 0-2 datapoints. This is detailed in the Materials and Methods, and we have amended Figure 1 to clarify this point.

1. Have a single timeline showing all 5 FFA groups, with a clear indication of how long before conditioning saline/ CPP was injected, and when the brains were micro-dissected (e.g. 2 hours). We have amended Figure 2a to incorporate this helpful suggestion.

2. Revise strong claims and speculations – such as NMDA-receptor activity controls the generation of these FFAs (page 9), can be re-written as ‘associated’ not controlled (if the statistics hold up), and the effect does not seem limited to associative learning but stimulation (CPP also changes unpaired groups).

Mention of beta-oxidation is also out of the blue (page 5). Instead, the various changes in different brain regions should be discussed with a circuitry perspective.

We have removed mention of beta-oxidation of octanoic acid. We have also modified our claims in the manuscript such that more emphasis is placed on correlation rather than causation.

Minor

issues:

1. Change the label of forebrain ‘FB’ to prefrontal cortex (PFC) – it is clear that the FB region was quite specific to the medial prefrontal cortex.

We have changed the use of forebrain (FB) to prefrontal cortex (PFC) throughout the text and figures of the manuscript.

2. Figure legend for each color plot should also state the name of the FFA (e.g., Palmitic (C 16:0)). It is difficult to understand the stacked color bar graphs based on the legend's use of the numeric short-hand nomenclature (e.g. 16:0) for the FFAs.

We have chosen "shorthand" nomenclature which is the most accurate way of describing the FFA species, in particular unsaturated FFAs. Medium chain unsaturated FFAs such as C18:1 have positional isomers (oleic acid, vaccenic acid, elaidic acid) which are not resolvable without using MS instrumentation which can fragment acyl chain double bonds.

3. In text, only use the names of acid. Sometimes both the name and number are used, and sometimes just a number is used. With the revised legend (comment above), it should be easy to understand if only the names are used in text.

We have amended FFA references in the text, such that the first mention of the FFA contains its name and carbon:double bond number. Subsequent references to the FFA contain just the name.

4. For the highlighted FFAs in text, state the corresponding % change – it has been done for some, but not all.

We have changed our data presentation such that % change is no longer an issue (see response to reviewer 1, points 2-5).

5. Abbreviations in general are inconsistent, some undefined, and some defined but used in a full form. A thorough proof-reading is necessary.

We have corrected this and apologise for the previous inconsistencies.

6. References to the corresponding author's cell culture study (Narayana et al., 2015) would benefit from specifying the species of cells (rats as in this study).

We have included this information in the manuscript.

7. Fig 1. missing bregma AP location for the cerebellum

We have now included this information in the manuscript.

8. Panel c of a few figures are wrongly introduced, as paired comparison, when no paired groups are present in some figures.

We have changed our data presentation such that this is no longer an issue (see response to reviewer 1, points 2-5).

9. Replace the misleading word 'Absolute response' to 'difference score'. When correctly presenting data with all 5 groups (6-7 including the requested controls), difference score can be calculated from the baseline group.

We have changed our data presentation such that this is no longer an issue (see response to reviewer 1, points 2-5).

Reviewer #4 (Remarks to the Author):

There are well established methods to fix the brain in vivo (See review Murphy EJ 2010) that have been established for over 30 years to measure brain FFA. Unfortunately, the ischemia with a 10 second delay can dramatically alter brain FFA, let alone 10 minutes. However, even

considering that, the FFA values presented in this study are higher than what is often reported for the total pool. In short the numbers are about 10000 times off what is accepted by the field. We thank the reviewer for spotting this very important issue. As explained above, we used this as impetus to re-evaluate the calibration and quantification of all the FFA data used in this study, and found and corrected systematic errors introduced during acquisition of the data. Upon reprocessing the data, we now present FFA concentrations in the pmol/mg of tissue range, in line with the literature. Note that the essential message of our manuscript has not been changed, as only the absolute values were inflated by these errors.

As biomedical researchers, we are bound by institutional animal ethics concerns, which have unfortunately precluded us from using focused microwave culling of animals. One of the issues raised in Murphy 2010 is that CO₂ anaesthesia and slow freezing on dry ice have an impact on prostaglandin levels. Neither of those techniques were used in our study. It is also worth noting that in a related study¹⁵, the time taken to remove the brain is an essential factor in changes of FFA level: at the 1 min time-point, decapitation with or without microwave does not make a difference. At 6 min the relative levels are comparable (one exception) between focused microwave and decapitation.

Ice cold perfusion and rapid snap freezing/dissection is extremely widely used, and implemented as a way to minimise the potential issues with ischaemic and post-mortem phospholipid metabolism. Furthermore, the results obtained in our study are comparable to those of other studies¹⁶.

Lastly, in Murphy (2010) the alternative technique suggested is to use parallel cohorts of animals, which is what we elected to do in this study. As the ice-cold perfusion was carried out identically in all groups, any post-mortem effects should have remained consistent across the study. We have now mentioned in the manuscript that our perfusion/dissection protocol was carried out in such a manner to minimise post-ischemic and post-mortem changes, in lieu of microwave-based techniques.

References

- 1 Rogan, M. T., Staubli, U. V. & LeDoux, J. E. Fear conditioning induces associative long-term potentiation in the amygdala. *Nature* **390**, 604-607, doi:10.1038/37601 (1997).
- 2 McKernan, M. G. & Shinnick-Gallagher, P. Fear conditioning induces a lasting potentiation of synaptic currents in vitro. *Nature* **390**, 607-611, doi:10.1038/37605 (1997).
- 3 Blanchard, R. J. & Blanchard, D. C. Crouching as an index of fear. *J Comp Physiol Psychol* **67**, 370-375, doi:10.1037/h0026779 (1969).
- 4 Maren, S. Long-term potentiation in the amygdala: a mechanism for emotional learning and memory. *Trends Neurosci* **22**, 561-567, doi:10.1016/s0166-2236(99)01465-4 (1999).
- 5 McDonald, R. J. *et al.* NMDA-receptor blockade by CPP impairs post-training consolidation of a rapidly acquired spatial representation in rat hippocampus. *Eur J Neurosci* **22**, 1201-1213, doi:10.1111/j.1460-9568.2005.04272.x (2005).
- 6 Rigoni, M. *et al.* Equivalent effects of snake PLA2 neurotoxins and lysophospholipid-fatty acid mixtures. *Science* **310**, 1678-1680, doi:10.1126/science.1120640 (2005).
- 7 Balleine, B. W. Neural bases of food-seeking: affect, arousal and reward in corticostriatolimbic circuits. *Physiol Behav* **86**, 717-730, doi:10.1016/j.physbeh.2005.08.061 (2005).
- 8 Bertran-Gonzalez, J., Laurent, V., Chieng, B. C., Christie, M. J. & Balleine, B. W. Learning-related translocation of delta-opioid receptors on ventral striatal cholinergic interneurons mediates choice between goal-directed actions. *J Neurosci* **33**, 16060-16071, doi:10.1523/JNEUROSCI.1927-13.2013 (2013).
- 9 Lynch, M. A. Long-term potentiation and memory. *Physiol Rev* **84**, 87-136, doi:10.1152/physrev.00014.2003 (2004).
- 10 Baez, M. V., Cercato, M. C. & Jerusalinsky, D. A. NMDA Receptor Subunits Change after Synaptic Plasticity Induction and Learning and Memory Acquisition. *Neural Plast* **2018**, 5093048, doi:10.1155/2018/5093048 (2018).
- 11 Franchini, L. *et al.* Linking NMDA Receptor Synaptic Retention to Synaptic Plasticity and Cognition. *iScience* **19**, 927-939, doi:10.1016/j.isci.2019.08.036 (2019).
- 12 Olvera, M. J. & Miranda, M. I. Specific inter-stimulus interval effect of NMDA receptor activation in the insular cortex during conditioned taste aversion. *Neurobiol Learn Mem* **164**, 107043, doi:10.1016/j.nlm.2019.107043 (2019).
- 13 Goodman, J., Hsu, E. & Packard, M. G. NMDA receptors in the basolateral amygdala mediate acquisition and extinction of an amphetamine conditioned place preference. *Behav Neurosci* **133**, 428-436, doi:10.1037/bne0000323 (2019).
- 14 Kim, J. J., DeCola, J. P., Landeira-Fernandez, J. & Fanselow, M. S. N-methyl-D-aspartate receptor antagonist APV blocks acquisition but not expression of fear conditioning. *Behav Neurosci* **105**, 126-133, doi:10.1037//0735-7044.105.1.126 (1991).
- 15 Cenedella, R. J., Galli, C. & Paoletti, R. Brain free fatty levels in rats sacrificed by decapitation versus focused microwave irradiation. *Lipids* **10**, 290-293, doi:10.1007/BF02532702 (1975).
- 16 Hennebelle, M. *et al.* Brain oxylipin concentrations following hypercapnia/ischemia: effects of brain dissection and dissection time. *J Lipid Res* **60**, 671-682, doi:10.1194/jlr.D084228 (2019).

Reviewers' comments:

Reviewer #3 (Remarks to the Author):

The authors did well in addressing most of the comments. Although the causal role of FFA in AFC is still unclear, the authors made an excellent effort re-organising the manuscript and adding data to strongly evidence the association between FFA and AFC. A few things still need to be addressed before publication (they should not be too effortful):

Major:

1. Specify whether Figure 1 animals were naïve and separate from the rest of the study.
2. Methods state that in this study, male rats were 6-8 weeks of age – which is adolescence to late adolescence in rats (Bell 2018; Madsen and Kim 2016), especially in males in which average puberty onset is P45 (Drzewiecki et al. 2016; Sellinger et al. 2020). Highlight their adolescence in the general discussion and discuss whether such strong response in the amygdala compared to the prefrontal cortex may indicate extinction-resistant fear memory in adolescence (Kim et al. 2011).

Minor:

3. State in the methods where the animals were acquired from.
4. Fig 1a: swap the placement of sections showing DH, so that the sections are shown in caudal to rostral arrangement.

References in this review:

- Bell, Margaret R (2018), 'Comparing Postnatal Development of Gonadal Hormones and Associated Social Behaviors in Rats, Mice, and Humans.', *Endocrinology*, 159 (7), 2596-613.
- Drzewiecki, C. M., Willing, J., and Juraska, J. M. (2016), 'Synaptic number changes in the medial prefrontal cortex across adolescence in male and female rats: A role for pubertal onset', *Synapse*, 70 (9), 361-8.
- Kim, J. H., Li, S., and Richardson, R. (2011), 'Immunohistochemical analyses of long-term extinction of conditioned fear in adolescent rats', *Cereb Cortex*, 21 (3), 530-8.
- Madsen, H. B. and Kim, J. H. (2016), 'Ontogeny of memory: An update on 40 years of work on infantile amnesia', *Behav Brain Res*, 298 (Pt A), 4-14.
- Sellinger, E. P., et al. (2020), 'Behavioral effects in adult rats exposed to low doses of a phthalate mixture during the perinatal or adolescent period', *Neurotoxicol Teratol*, 79, 106886.

Reviewer #4 (Remarks to the Author):

The authors have now reprocessed their FFA data and corrected a systematic error that led to levels being several thousand-fold higher than the literature. The authors state that their brain FFA values now reflect citation 16 Henebelle et al., Unfortunately Henebelle et al., did not measure FFA in their study, but rather used an LC/MS approach to measure oxylipins derived from PUFA. Thus, it is not clear why the authors have made this statement. What citation 16 shows is that rapid artifacts are produced in non-microwaved brains with levels of oxylipins changing dramatically within 10s of seconds for tissue dissection time. This is important later.

The authors raise ethical questions as to why they cannot use microwave fixation stating: “As biomedical researchers, we are bound by institutional animal ethics concerns, which have

unfortunately precluded us from using focused microwave culling of animals.” Yet the School of Biomedical Sciences Integrated Physiology Facility at their university lists a microwave fixation system on its core facility website. ” <https://biomedical-sciences.uq.edu.au/facilities/integrated-physiology-facility> that is routinely used by faculty there <https://www.ncbi.nlm.nih.gov/pmc/articles/PMC3705438/>

Nevertheless, the authors used a reasonable approach with cold perfusion and make comparisons to a control. However, the problem this creates is data like extended figure 2 (bottom panel) how did the authors take out 20+ brain regions (figure 1 I has 6) at the same time? The details in the paper do not appear sufficient to make these types of measurements as the dissection took 10 minutes. Labs that do this use a stopwatch. It is not clear the authors used an appropriate methodology to compare between brain regions. Even phospholipid levels change after such long times (Trepanier et al., 2017 and Bazan 1970)

Reviewers' comments:

Reviewer #3 (Remarks to the Author):

The authors did well in addressing most of the comments. Although the causal role of FFA in AFC is still unclear, the authors made an excellent effort re-organising the manuscript and adding data to strongly evidence the association between FFA and AFC. A few things still need to be addressed before publication (they should not be too effortful):

We thank the reviewer for his/her encouraging remarks. We have addressed these issues as follows:

Major:

1. Specify whether Figure 1 animals were naïve and separate from the rest of the study.

We have now clearly stated in the figure legend that the data was from the control saline unpaired animal from our AFC experiments.

2. Methods state that in this study, male rats were 6-8 weeks of age – which is adolescence to late adolescence in rats (Bell 2018; Madsen and Kim 2016), especially in males in which average puberty onset is P45 (Drzewiecki et al. 2016; Sellinger et al. 2020). Highlight their adolescence in the general discussion and discuss whether such strong response in the amygdala compared to the prefrontal cortex may indicate extinction-resistant fear memory in adolescence (Kim et al. 2011).

This has now been fixed through the addition of a paragraph stating that the rats were adolescent and therefore their amygdala response was likely to be heightened. Our additional text (in the result section) reads as follows (and include references suggested by reviewer):

“It is worth noting that the 6-8 wk animals used in this study represent mid-late adolescent males¹⁻⁴. Heightened emotional responses to negative events during adolescence are well documented, and adolescent rats have been demonstrated to show dramatically reduced extinction of fear-induced freezing behaviour compared to adults⁵. The heightened FFA response in the amygdala observed in this study is likely to at least partly reflect this developmental process.”

Minor:

3. State in the methods where the animals were acquired from.

We have updated the Methods with the following sentence:

“Male Sprague-Dawley rats (380 ± 15 g, 6 – 8 weeks of age) were obtained from the University of Queensland Biological Resources (UQBR) facility.”

4. Fig 1a: swap the placement of sections showing DH, so that the sections are shown in caudal to rostral arrangement.

We have amended the Figure accordingly.

Reviewer #4 (Remarks to the Author):

The authors have now reprocessed their FFA data and corrected a systematic error that led to levels being several thousand-fold higher than the literature. The authors state that their brain FFA values now reflect citation 16 Henebelle et al., Unfortunately Henebelle et al., did not measure FFA in their study, but rather used an LC/MS approach to measure oxylipins derived from PUFA. Thus, it is not clear why the authors have made this statement. What citation 16

shows is that rapid artifacts are produced in non-microwaved brains with levels of oxylipins changing dramatically within 10s of seconds for tissue dissection time. This is important later.

We apologize for the mistake in the citation. We have now included a more appropriate reference for the FFA levels in the main text. It reads as follows:

“We took advantage of the nanomolar sensitivity of FFAST to quantitatively survey the FFA landscape (18 targeted FFA species, Supp. Table 1) across 6 regions of the rat brain, reporting FFA abundances in the pmol/mg wet weight range consistent with previous literature⁴⁰.”

The authors raise ethical questions as to why they cannot use microwave fixation stating: “As biomedical researchers, we are bound by institutional animal ethics concerns, which have unfortunately precluded us from using focused microwave culling of animals.” Yet the School of Biomedical Sciences Integrated Physiology Facility at their university lists a microwave fixation system on its core facility website. ” <https://biomedical-sciences.uq.edu.au/facilities/integrated-physiology-facility> that is routinely used by faculty there <https://www.ncbi.nlm.nih.gov/pmc/articles/PMC3705438/>

We appreciate the concern of the reviewer. We were not aware of this facility on the UQ campus and, for this reason, did not include its use in our animal ethics dealing with this particular project.

Nevertheless, the authors used a reasonable approach with cold perfusion and make comparisons to a control. However, the problem this creates is data like extended figure 2 (bottom panel) how did the authors take out 20+ brain regions (figure 1 I has 6) at the same time? The details in the paper do not appear sufficient to make these types of measurements as the dissection took 10 minutes. Labs that do this use a stopwatch. It is not clear the authors used an appropriate methodology to compare between brain regions. Even phospholipid levels change after such long times (Trepanier et al., 2017 and Bazan 1970)

We could not agree more with the reviewer. In fact, this comment made us realize how inadequate the description of the methods was, for which we profusely apologize.

We have now completely rewritten this section based on the precise timing of our handling of the brain dissection which goes as follows:

“Brain tissue was collected under conditions designed to minimize ischaemic and post-mortem lipid metabolism. Animals were deeply anaesthetized with isoflurane and transcardially perfused with ice-cold oxygenated artificial cerebrospinal fluid (ACSF) containing 88 mM NaCl, 2.5 mM KCl, 25 mM NaHCO₃, 10 mM D-glucose, 1.2 mM NaH₂PO₄, 1.3 mM MgCl₂ and 2.5 mM CaCl₂ to rapidly remove all blood and to chill the brain. The animals were quickly decapitated and their brains were extracted, snap-frozen in liquid nitrogen and stored at -80°C. In a coldroom, each frozen brain was promptly transferred to the mounting block of a cryostat (NX70, Thermo Fisher Scientific) operating at -20°C, and 80-100 µm thick sections were cut and transferred to glass slides within the cold cryostat. Slides were quickly transferred to a microscope (Olympus SZ51) next to the cryostat, and appropriate brain regions were rapidly dissected from each frozen slice and placed into 2ml tubes (Eppendorf) on dry ice. The dissection of specific regions from a given slice took less than a minute. On average, 70 slices were cut for each brain. Each tube, representing a specific brain region, contained material dissected from approximately 20 sequential brain slices. All samples were stored at -80°C until subsequent lipid extraction.

Extraction procedures were also performed under cold conditions, and at no stage prior to this were brain samples allowed to thaw.”

We are very sorry that our previous description has led the reviewer to rightly question our results based on perceived lengthy dissection of the brain at room temperature. We hope that this much more accurate description of our handling of the brain dissection clarifies our rationale and methodology. Key to the rationale was that the tissue was frozen as soon as it was taken out of the animal and was kept frozen until the lipid extraction. We appreciate the detailed feedback at multiple iterations of our manuscript which has greatly improved its quality and avoided a potentially embarrassing post-publication correction.

Reviewers' comments:

Reviewer #3 (Remarks to the Author):

I have already endorsed this manuscript at the last revision.